# LEARNING TO MODEL EDITING PROCESSES

## ABSTRACT

Most existing sequence generation models produce outputs in one pass, usually left-to-right. However, this is in contrast with a more natural approach that humans use in generating content; iterative refinement and editing. Recent work has introduced edit-based models for various tasks (such as neural machine translation and text style transfer), but these generally model a *single* edit. In this work, we propose *modeling editing processes*, modeling the whole process of iteratively generating sequences. We form a conceptual framework to describe the likelihood of multi-step edits, and describe neural models that can learn a generative model of sequences based on these multi-step edits. We introduce baseline results and metrics on this task, finding that modeling editing processes improves performance on a variety of axes on both our proposed task and related downstream tasks compared to previous single-step models of edits.[1]

## 1 INTRODUCTION

Revising and editing are a central part of the the human creative worflow, with most original content (e.g. art, books, articles, source code) being developed not in a single iteration, but in many iterations with each more refined than the last. How can we model these *editing processes* from inception to completion? In this paper, we attempt to provide a first answer to this question, specifically focusing on generation of sequential data such as natural language documents or source code.

Most current work on language generation tasks such as machine translation (Vaswani et al., 2017), language modeling (Baevski & Auli, 2018), or summarization (See et al., 2017) generates the target sentence or document in a single pass (usually from left to right). There has been a reasonable amount of work that can generate edits to existing sequences for the purposes of post-editing, grammatical error correction (Omelianchuk et al., 2020), text style transfer (Mallinson et al., 2020; Malmi et al., 2020; Reid & Zhong, 2021), sentence fusion (Malmi et al., 2019), or machine translation (Gu et al., 2019). However, these works all 1) model only a single editing step and 2) do not fully define a model of incrementally editing a document from a blank slate to the final text, and thus do not stand in for the one-pass generative models of sequences described above.

In this context, we propose the task of *modeling editing processes* in which we look to explicitly model the likelihood of the entire process of revising a document to a polished form. In particular, and in contrast to previous works on modeling edits, we hypothesize that in order to edit more accurately, instead of simply learning to predict the next revision given the current revision, we should have context of multiple previous revisions when deciding when and how to edit the document next. Given the novelty of this task, this paper simultaneously proposes 1) both baseline and novel models for the task, 2) evaluation methodology, and 3) evaluation datasets that can be used to compare model efficacy.

The proposed multi-step editing model predicts discrete edit operations (Levenshtein, 1966) to enable progressive refinement as shown in Figure 1, rather than framing sequence editing as a sequence to sequence task (Reid & Zhong, 2021; Faltings et al., 2021). In the figure, for each step of the editing process discrete operations (insert, replace, delete, keep) are predicted and then actions (such as generating a replaced span) are performed based on this. This has two benefits: 1) it allows the model to scale well with respect to input sequence length, and 2) allows us to make substantial changes with fewer actions (Grangier & Auli, 2018). We use these edit operations to condition a semi-autoregressive model that is able to insert and replace multiple spans at once. Combined with

---

[1]All data and code will be open-sourced upon acceptance.

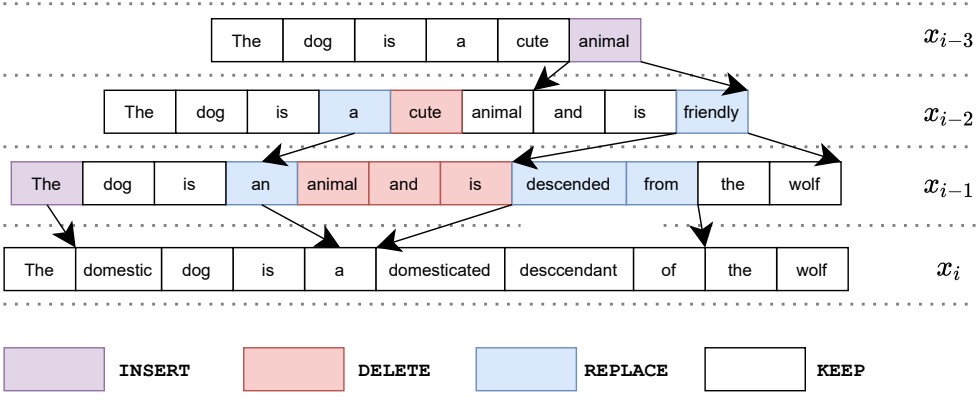

Figure 1: An example of a natural editing process based on the description of "Dog" on Wikipedia. The legend below denotes the edit operations for each step of this process.

an encoder that is able to quickly specify which spans of text need to be changed and *how*, this allows for considerable changes to be made to the text (including insertion, deletion, re-ordering, and replacement) in a relatively simple and cheap manner. Furthermore, this allows us to disentangle how likely the model is to operate (replace, delete, etc...) on a given span, and how likely the model thinks the generated text for a given span is. As we are modeling editing *processes*, and hypothesize that context from edits applied to the sequence are helpful, we propose a method for edit-aware sequence compression which can compress sequences into their edit operations and use *relative edit positional embeddings* to specify the position of edits relative to each other.

To evaluate the efficacy of proposed and baseline models, we propose an evaluation metric, *edit perplexity* (ePPL), which combines both of these ideas for a unified metric to measure edit likelihood. However we can easily disentangle this into *operation perplexity* (oPPL) and *generative perplexity* (gPPL) which measures the likelihood of performing a given operation on a span, versus the likelihood of newly generated text for the next revision.

Given that the task of modeling natural editing processes in itself is novel, we collect new datasets to study this behavior; WIKIREVISIONS and CODEREVISIONS. These datasets, in the code and natural language domains respectively, cover over 2.5M and 2.3M natural sequential revisions.

We train and evaluate our proposed models on these datasets. Our experimental results demonstrate that edit perplexity, generative perplexity, and operation perplexity are all reduced when one models edits with previous changes/edits as context. We also qualitatively demonstrate the ability of the model to generate natural edit sequences, and demonstrate the utility of the learned representations on downstream tasks of commit message generation (Loyola et al., 2017) and edit intention classification (Yang et al., 2017).

## 2 PROBLEM DEFINITION

Let $X = \{\boldsymbol{x}_0, \boldsymbol{x}_1, \ldots, \boldsymbol{x}_N\}$ be a series of $N$ versions of a document, where the $i$th revised document is denoted by $\boldsymbol{x}_i$. $\boldsymbol{x}_0$ represents an initial state (generally the null string), and $\boldsymbol{x}_N$ represents the current state of the edited document. The probability of this series of document versions occurring can be decomposed as

$$p(X) = \prod_{i=1}^{N} p(\boldsymbol{x}_i | \boldsymbol{x}_0^{i-1}), \tag{1}$$

where $\boldsymbol{x}_0^{i-1} := \boldsymbol{x}_0, \ldots, \boldsymbol{x}_{i-1}$ (similarly below). The right hand side is the likelihood of the transformation of the previous document version $\boldsymbol{x}_{i-1}$ to the current document version $\boldsymbol{x}_i$ given the previous revision history $\boldsymbol{x}_{<i}$. We refer to the likelihood of the whole revision process as the *edit likelihood*, and judge learned models based on their ability to achieve high edit likelihood on held-out data.

Note that standard generative models (specifically language models; LMs) calculate the probability of only the final version $p(\boldsymbol{x}_N)$, whereas the proposed formulation calculates the probability of the entire sequence of document edits. It nonetheless could be used to calculate the final version's likelihood by treating the editing process as latent and marginalizing over it

$$p(\boldsymbol{x}_N) = \sum_{\{\tilde{X}=\tilde{\boldsymbol{x}}_1^N | \tilde{\boldsymbol{x}}_N = \boldsymbol{x}_N\}} p(\tilde{X}). \qquad (2)$$

Thus our formulation, in contrast to previous single-step models of edits (Yin et al., 2019b; Malmi et al., 2019; Reid & Zhong, 2021), can also be used to define a generative model over single documents. It is also worth noting that the final document likelihood is lower-bounded by the edit likelihood; i.e. $p(\boldsymbol{x}_N) \geq p(X)$.

## 3 MODELING EDITING PROCESSES

In this section, we now describe our approach to actually modeling these sequences of edits through (1) a decomposition of the modeling process into a sequential process of modeling edit operations then actual edits, and (2) neural model of modeling these operations and edits.

### 3.1 MODELING OPERATIONS AND OPERATION-CONDITIONED EDITS

While the probability $p(\boldsymbol{x}_i | \boldsymbol{x}_0^{i-1})$ of the next document given all previous document versions could theoretically be modeled with a single neural sequence model, this is infeasible computationally (and likely infeasible from learning perspective as well). To simplify this problem, we employ the $n$-th order Markov assumption, assuming that the probability of the next document is conditioned only on the previous $n$ documents $p(\boldsymbol{x}_i | \boldsymbol{x}_{i-n}^{i-1})$. This probability could be modeled directly, and in fact in the case of $n = 1$ this becomes analogous to the single-step editing problem tackled by previous work (Yin et al., 2019b; Malmi et al., 2019; Reid & Zhong, 2021; Faltings et al., 2021). To our knowledge, no previous work has modeled natural editing processes with $n > 1$.

However, in the interest of both efficiency and efficacy, we take an alternative approach where we first predict a set of edit operations $\mathbf{e}_i$, and then predict the next document version based on the previous documents and these edit operations:

$$p(\mathbf{x}_i | \mathbf{x}_{i-n}^{i-1}) \approx p(\mathbf{x}_i, \mathbf{e}_i | \mathbf{x}_{i-n}^{i-1}) \qquad (3)$$

$$= p(\mathbf{x}_i | \mathbf{e}_i, \mathbf{x}_{i-n}^{i-1}) p(\mathbf{e}_i | \mathbf{x}_{i-n}^{i-1}). \qquad (4)$$

The first approximation becomes an equality when the edit operations can be deterministically derived from $\mathbf{x}_i$ and $\mathbf{x}_{i-1}$, i.e. $p(\mathbf{e}_i | \mathbf{x}_i, \mathbf{x}_{i-1}) = 1$, as is the case described below.

**Edit Operations.** We base the edit operations in $\mathbf{e}$ on those calculated by the Levenshtein algorithm (Levenshtein, 1966), including token-level insertions, deletions, and substitutions. These are expressed as four operations insert, delete, keep, and replace denoted by {INSERT, DELETE, KEEP, REPLACE}. For multi-word insertions and replacements, e.g. a replacement of a contiguous span of words, we apply the the same REPLACE label to all tokens in this span. An example of each operation is shown in Figure 1.

**Decomposed Edit Likelihood.** We can then re-define our previous formulation of edit likelihood:

$$P(\mathbf{x}_1^N) = \prod_{i=1}^{N} p(\mathbf{x}_i | \mathbf{e}_i, \mathbf{x}_{i-n}^{i-1}) p(\mathbf{e}_i | \mathbf{x}_{i-n}^{i-1}), \qquad (5)$$

and analogously define edit log-likelihood

$$\mathcal{L}_{\mathbf{xe}} := \log P(\mathbf{x}_1^N) = \sum_{i=1}^{N} \log p(\mathbf{x}_i | \mathbf{e}_i, \mathbf{x}_{i-n}^{i-1}) + \log p(\mathbf{e}_i | \mathbf{x}_{i-n}^{i-1}). \qquad (6)$$

We can further decompose this into only the components corresponding to the edit operations $\mathcal{L}_{\mathbf{e}} := \sum_{i=1}^{N} \log p(\mathbf{e}_i | \mathbf{x}_{i-n}^{i-1})$, or the operation-conditioned edits $\mathcal{L}_{\mathbf{x}|\mathbf{e}} := \sum_{i=1}^{N} \log p(\mathbf{x}_i | \mathbf{e}_i, \mathbf{x}_{i-n}^{i-1})$, both of which we will utilize for devising evaluation metrics in Section 5.2 below.

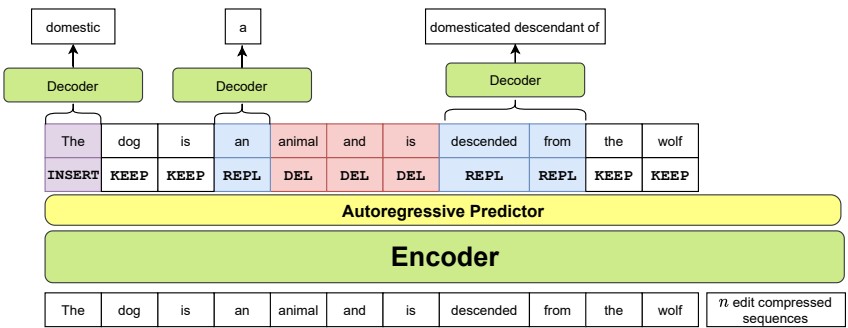

Figure 2: EDITPRO given the examples of modeling $p(\boldsymbol{x}_i|\boldsymbol{x}_{i-1})$ from Figure 1. We feed the input tokens into an encoder with an autoregressive tag predictor, and then use the predicted edit operations to condition the generation of `REPLACE` and `INSERT` spans.

## 3.2 EDITPRO

In this section, we propose a model of multi-step editing processes, EDITPRO, which is based on a semi-autoregressive edit-conditioned encoder-decoder model with a Transformer (Vaswani et al., 2017). The model (depicted in Figure 2) contains three main components: (1) an edit encoder, (2) an operation classifier and (3) an insertion-replacement decoder.

**Edit Encoder.** The encoder $f_{\text{enc}}$ takes in a document version $\boldsymbol{x}_{i-1}$ and feeds it through multiple self-attention and feedforward layers (Vaswani et al., 2017) to produce contextual representations for each token. In the case that we perform variable-order edit modeling, we use cross-attention to feed in representations of previous edit steps. For models where $n > 1$, we feed in $n-1$ additional edit sequences – we describe this process after describing our methods for edit sequence prediction.

**Edit Operation Prediction.** We use an autoregressive tagger, using a single Transformer layer with a causal attention mask, that models the probability of each edit in edit operation sequence $\mathbf{e} = e_1^M$ from left to right, $p(e_j|e_1^{j-1})$. Notably, we also performed preliminary experiments with a tagger that predicts operations independently, but found it was heavily biased towards the `KEEP` operation as most words are kept in any single document revision, and thus did not produce coherent multi-word edit sequences when sampling sequences of edits.

**Generating Replacements and Insertions.** When editing, given our four Levenshtein operations (`INSERT`, `REPLACE`, `KEEP`, `DELETE`), two of them — `INSERT` and `REPLACE` — entail generation of new content conditioned on the current revision of the document. Given our predicted edit operations $e$, we propose a semi-autoregressive model with a Transformer decoder that can decode multiple spans in parallel for efficiency purposes. Each edit span contains the following properties: it has a start index (denoted by $s_{\text{start}}$), end index (denoted by $s_{\text{end}}$), and an operation type (denoted by $s_{\text{type}}$) . Note that these can be simply be extracted by looking at contiguous spans of a certain type in an edit (e.g. `REPLACE` for *descended from* → *domesticated descendant of* in Figure 1). We use a mean pooling operation to aggregate the contextual vectors produced by $f_{\text{enc}}(\boldsymbol{x})$ into span representation $\hat{x}_s$ :

$$\hat{x}_s = \frac{1}{s_{\text{end}} - s_{\text{start}}} \sum_{t=s_{\text{start}}}^{s_{\text{end}}} f_{\text{enc}}(\boldsymbol{x})_t \tag{7}$$

We then update the span representation $\hat{x}_s$ by taking the sum of the appropriate operation embedding for the span type and the current span representation and feed it to a multi-layer perceptron with an intermediate non-linearity: $\hat{x}_s \leftarrow \text{MLP}(W_{\text{op}}(\boldsymbol{e})_s + \hat{x}_s)$, where $W_{\text{op}}$ denotes an embedding matrix for each operation. $\hat{x}_s$ is then used to initialize the `` token for the decoder span to further condition the generative process.

**Encoding Edit History.** As we look to investigate variable order edit modeling over long sequences of text, we need a way to be able to represent edits in a way useful for predicting the next editing steps. Previous work (Yin et al., 2019a; Marrese-Taylor et al., 2021; Yao et al., 2021) has focused largely on learning a single-vector representation for edits which is compressed but limited in expres-

| Dataset | Num. Edits | Avg. Len (Max/Min) | % Keep | % Insert | % Replace | % Delete |
|---------|-----------|--------------------|--------|----------|-----------|----------|
| WIKIREVISIONS | 2.5M | 333 (9992/1) | 82.4% | 0.1% | 8.7% | 8.8% |
| CODEREVISIONS | 2.3M | 774 (9725/1) | 76.9% | 0.1% | 11.3% | 11.7% |

Table 1: Dataset statistics on CODEREVISIONS and WIKIREVISIONS, average length is measured by whitespace tokenization

siveness. One the other hand, a perhaps more intuitive way taken from common Transformer-based (Vaswani et al., 2017) models would be to use cross-attention between all $n$ previous documents, which is more expressive but prohibitively expensive when $n$ is scaled upwards.

Instead, we make a compromise between the above approaches, leveraging predicted edits $\mathbf{e}_{i-n}^{i-1}$ to compress the sequence and their derived spans (as discussed above). Given each of these spans, we compute the edit-compressed sequence, composed of a sequence of vector representations with each vector representing a different span. For each span in each of the previous revisions in $\boldsymbol{x}_{i-n}^{i-1}$), we mean pool the encoder (pre-edit) and the decoder (post-edit) representations for that span. We then sum this representation with the operation representing its edit operation and feed it into an MLP. Once we have done this for each span, we sum a learned *relative edit positional embedding*, where we learn an embedding matrix where each index in the matrix represents each order. This is to specify the order of the previous edits . Finally, we compose these into a sequence and treat that as the "edit-compressed" sequence representation for that edit.

**Turning Pre-trained Encoder-Decoder Models into Editors.** Despite the fact that our model introduces both an edit prediction and a semi-autoregressive component, it is easy to finetune a pretrained language model into an editor with our method as it uses vanilla Transformer layers as a backbone. We perform this by batching various spans and their conditioning variables together and training the model to adapt to decode these in parallel.

## 4 DATA

While some datasets of edits exist (Faruqui et al., 2018; Marrese-Taylor et al., 2021), to our knowledge they only consider a single isolated edit, i.e. dealing with a document $X = \{\boldsymbol{x}_0, \boldsymbol{x}_1\}, N = 1$. As we propose learning to model multi-step edits, we develop new datasets in both the code and natural language domains. In addition, previous datasets have only concerned themselves with *atomic* edits (Faruqui et al., 2018) which only occur at a small scale (usually sentence-level), and we instead look to model larger-scale edits as document level changes, which are more representative of the natural editing process.

### 4.1 WIKIREVISIONS

For the natural language domain of Wikipedia, we collect data for each revision using dumps from English Wikipedia. Given that the dumps are provided in the XML format, we extract the text with `beautifulsoup` and remove wikitext (custom Wikipedia markup) with `wikiextractor`. With this sanitized data, we gather revision of each document in chronological order removing any metadata-based edits which were stripped as a result of the sanitization process. Now, with our sets of revisions we tokenize all text with the sentencepiece model used by Radford et al. (2018); Liu et al. (2019) for congruence with pre-trained models (see Section 3.2). We pre-compute Levenshtein operations using `python-Levenshtein`[2] for use during training. In the case that an article exceeds 2000 tokens, we split the articles into its subsections and treat each subsection as an article (for the purpose of modeling editing processes). Dataset statistics are shown in Table 1. We note that there is a significant imbalance for the `INSERT` operation, this is because we define insertions to be applied to the token preceding the insertion (as shown in Figure 1), rather than applied to an entire span (as we do for the deletion, replacement, and keep operations).

**Edit Summaries.** When extracting each edit we keep the edit summary (akin to a commit message) supplied by the editor at time of editing. We then curate these comments and develop a dataset

---

[2]`https://pypi.org/project/python-Levenshtein/`

for usage on downstream tasks—for both edit summary generation (Loyola et al., 2017) and edit-summary-conditioned text editing (Faltings et al., 2021).

## 4.2 CODEREVISIONS

When buildling CODEREVISIONS, we scrape a total of 700 Python GitHub repositories using the MIT License with at least 1000 commits and 500 stars. We extract line-level patches from each repository's commit history when forming our code-based corpus and progressively apply each patch and compute the token-level Levenshtien operations between each revision. Note that we also keep commit messages for each commit. For this dataset we operate on the file level. For each series of revisions, we precompute Levenshtein operations based on tokens derived from a `sentencepiece` (Kudo & Richardson, 2018) model with a 10k vocabulary. We also curate a dataset of revisions with commit messages as described in the previous subsection.

## 5 EXPERIMENTAL SETUP

### 5.1 BASELINES

We use the following baseline for our edit modeling task: (1) LEWIS (Reid & Zhong, 2021), a state-of-the-art single-step editing model, which uses a separate encoder-only tagger and sequence-to-sequence generator setup during training.

### 5.2 METRICS

Many previous works on editing (Malmi et al., 2019; Gu et al., 2019; Reid & Zhong, 2021) have used non-likelihood based metrics such as BLEU or F1, representing the downstream tasks which they were trained for, however as we look to model the intrinstic editing process (i.e. applying and generating a sequence of edits) and the likelihood of performing certain edits, we propose the following metrics. Note that $|\mathbf{x}|$ refers to the token count for the newly generated/inserted spans, and $|\mathbf{e}|$ refers to the number of edit operations:

**Edit Perplexity (ePPL)** is the exponent of the negative log likelihood for both the edit operations and generated outputs, divided by the length of both sequences, $\exp(\frac{-\mathcal{L}_{\mathbf{xe}}}{|\mathbf{x}|+|\mathbf{e}|})$.

**Generation Perplexity (gPPL)** measures the likelihood of generating replaced or inserted spans when compared with the ground truth edit sequence as follows $\exp(\frac{-\mathcal{L}_{\mathbf{x}|\mathbf{e}}}{|\mathbf{x}|})$.

**Operation Perplexity (oPPL)** $\exp(\frac{-\mathcal{L}_{\mathbf{e}}}{|\mathbf{e}|})$ is the likelihood of predicting a set of edit operations.

### 5.3 TRAINING SETUP

We train our models using the Transformer implementation in the HuggingFace (Wolf et al., 2020) library. We tokenize data using SentencePiece (Kudo & Richardson, 2018), using the same vocabulary employed in Liu et al. (2019); Lewis et al. (2019) for natural language, and using a custom 10k subword vocabulary for code. We use the Transformer architecture of a hidden dimension of 768, feed- forward size of 3072, and 6 layers for both the encoder and decoder. We initialize all natural language models with BART (Lewis et al., 2019), and code models with random initialization. We set the maximum sequence length for all models to be 2048, using a batch size of 65K tokens distributed over 8 A100 GPUs.

### 5.4 DOWNSTREAM TASKS

In addition to assessing our proposed model's generative capacity, we assess the quality of the learned representations on downstream tasks:

**Conditional Editing** We also continue training using the commit messages gathered during the cleaning process as a conditioning variable, essentially reformulating our $p(\boldsymbol{x}_i|\boldsymbol{x}_{i-n}^{i-1})$ to $p(\boldsymbol{x}_i|\boldsymbol{x}_{i-n}^{i-1}, \boldsymbol{c})$ to add the additional conditional variable $\boldsymbol{c}$, which we set to be the edit summary or commit message in this setting. With our model, we append the comment to each document , delimiting with a separator token `` as follows: `DOCUMENT  COMMENT`.

**Edit-conditioned Generation.** We define edit-conditioned generation to be tasks which rely on intermediate edit representations of documents to generate text describing the changes in text, similar to that proposed by Loyola et al. (2017) for source-code commit message generation. As we aim to determine whether the information contained about the edit itself is more informative as we add additional context, we condition the generation solely on the edit-compressed representations of the last edit step. To accomplish this, we use a randomly initialized Transformer decoder with cross-attention on these edit-compressed representations.

**Edit-conditioned Classification.** In the natural language domain, we also test our representations on an edit-oriented classification task, namely semantic intent classification (Yang et al., 2017). In Yang et al. (2017), they classify 5,777 Wikipedia revisions into 10 intention classes, such as "Clarification", "Vandalism", and others with each representing a different intention. We form splits of 4,601 train examples, 588 valid examples, and 588 test examples.[3] Similarly to our setup for edit-conditioned generation, we also test our classifier (consisting of a self-attentive span extractor (Lee et al., 2017) and a multi-layer perceptron) on this task.

## 6 RESULTS

### 6.1 EDIT MODELING

Results on edit modeling for both CODEREVISIONS and WIKIREVISIONS can be seen in Table 2, where we measure edit perplexity, operation perplexity, and generative perplexity. We first note that our model significantly outperforms LEWIS (Reid & Zhong, 2021) on WIKIREVISIONS, by 8.6 ePPL, suggesting that our model formulation is superior at this task. We believe that this stems from the fact that our model is trained to explicitly generate the newly added spans, and because it directly connects the operation prediction and generation processes. We also take note that ePPL decreases when the order of context increases. In particular, we take note of the significant gain when introducing the notion of editing processes to the model in our 2-order setting (in contrast to a single edit step), with a reduction of 3.4 ePPL on natural language and 4.4 ePPL on source code.

We also note that while the gPPL consistently decreases as the number of orders increases, operation perplexity does not perform as consistently. Interestingly, we find that single-order models tend to be more confident with respect to keeping tokens (the overwhemingly dominant edit operation), while other operations (deletions, replacements and insertions) are not predicted as well. In contrast, learning higher-order editing processes almost universally decreases the oPPL for non-KEEP operations, indicating the necessity to model longer context to capture these rarer edit operations.

***Likely*** and ***Unlikely*** **Edits.** We perform a qualitative analysis on a subsample 4,000 natural language edits,[4] examining which edits are judged to be likely (or unlikely) and with respect to which metrics. We do this by identifying outlier values for each metric (significantly above or below the average) and further analysing these for unique properties.

As a result, we found that many of the edits with higher oPPL were spam and vandalism-type edits, as many of the edit operations have more of a random nature. However we notice that generative perplexity was much lower as these edits tend to be repetitive in nature with the same ngrams often being repeated for long spans. However, we notice that, irrespective of the number of orders, when editing *reverted* spam-like content, the oPPL for the REPLACE and DELETE operations are extremely low (on average 1.07 and 4.4 respectively). The importance of variable-order modeling was particularly evident these revisions where the gPPL in the single-order setting averages at 123.90 gPPL, however when using 2-orders we are able to attain 67.83 gPPL indicating that the edit-compressed sequences provide useful context about the previous revisions. We also notice that models are able to predict insertions (2.25 INSERT oPPL) significantly better when they come after the end of a sentence, representative of many insertions in Wikipedia. We also notice that outside of the above settings, models with extra context generally predict more likely edits supporting the notion of modeling edit processes compared to modeling changes individually.

---

[3]We contacted the authors, but they were no longer able to release the folds they used in their K-fold cross-validation setup, hence our creation of splits

[4]A precursory examination found these to be more interpretable than code edits.

| DATASET | Model | ePPL | gPPL | oPPL | | | |
|---|---|---|---|---|---|---|---|
| | | | | DEL | KEEP | REPL | INS |
| **WIKIREVISIONS** | LEWIS | 65.94 | 48.85 | 24.29 | **1.09** | 19.49 | 507.76 |
| | EDITPRO (1-order) | 57.32 | 42.43 | 25.53 | **1.09** | 18.36 | 1826.21 |
| | EDITPRO (2-order) | 53.91 | 39.87 | 20.70 | 1.13 | 15.49 | 376.15 |
| | EDITPRO (3-order) | **50.84** | **37.66** | **19.30** | 1.13 | **14.88** | **252.14** |
| **CODEREVISIONS** | EDITPRO (1-order) | 34.22 | 28.02 | 125.21 | **1.05** | 10.38 | 544.57 |
| | EDITPRO (2-order) | 30.85 | 26.26 | 84.77 | **1.05** | 9.30 | **304.90** |
| | EDITPRO (3-order) | **29.47** | **25.37** | 75.19 | 1.06 | **8.16** | 441.42 |

Table 2: Results on Edit Modeling

| DATASET | Model | BLEU | F1 | ePPL ($\Delta$) |
|---|---|---|---|---|
| **WIKIREVISIONS** | EDITPRO (1-order) | 10.7 | 57.8 | 54.72 (-2.60) |
| | EDITPRO (2-order) | 11.3 | **61.3** | 51.83 (-2.08) |
| | EDITPRO (3-order) | **11.6** | 61.2 | 49.91 (-0.93) |
| **CODEREVISIONS** | EDITPRO (1-order) | 13.8 | — | 33.65 (-0.57) |
| | EDITPRO (2-order) | 14.3 | — | 30.13 (-0.72) |
| | EDITPRO (3-order) | **14.5** | — | **29.08** (-0.39) |

Table 3: Results on Edit Generation (BLEU), Edit Classification (measured with micro-F1), and Conditional Edit Generation (measured Edit Perplexity = ePPL). Note that the $\Delta$ symbol refers to the change between the model's non-message conditioned version in Table 2.

## 6.2 DOWNSTREAM PERFORMANCE

Results on conditional edit generation, edit classification and edit-conditioned generation can be seen in Table 3 . We note that findings generally follow the edit modeling results, with additional context improving performance further giving supporting evidence to modeling editing processes. Specifically, increasing context from single-order to 3-order improves commit message generation performance by 1.9 and 0.7 BLEU for both natural language and source code respectively. We also note that ePPL decreases similarly when we add natural language conditioning to the editing process, which indicates that multi-order editing encodes fine-grained information not found in the commit message.

**Edit Modeling** In particular, when performing editing using an editor pre-trained on edit modeling, we note that when sampling from the autoregressive tagger it almost always predicts KEEP with extremely high confidence , given the overwhelming class majority. We instead perform a variety of posterior regularisation (Ganchev et al.), reducing the probability of the KEEP class until the sampled edit labels to grow closer in proportion to the true distribution of edit operations (Table 1). Combined with this technique, we are able to generate more diverse edits, which we show in Table 4.

**Semantic Coherence** In looking at the example generations in Table 4 we note that the generated text is not perfectly semantically coherent, despite showing topical coherence and some degree of discourse coherence. We believe this is largely due to the size of the language model we use, being trained solely on Wikipedia data (which contains a variety of minor edits including spam/vandalism). Given this, we expect improved semantic coherence upon scaling up data, data diversity and model scale. However, we note the improved context-awareness of the edit path shown by the 2-order model over the 1-order model, providing qualitative evidence for modeling editing processes and looking at different forms of document construction.

## 7 RELATED WORK

**Learning Edit Representations.** Previous work on learning properties inherent to editing has largely focused on learning distributed representations of edits. Yin et al. (2019a) proposed this task, using an attentional sequence-to-sequence model to learn representations. Marrese-Taylor et al. (2021) expands upon this approach, introducing an auxiliary loss and a variational setup. More recently, Yao et al. (2021) worked on using this approach for editing tree-structured data. However, in

| | |
|---|---|
| Initial Sentence (1-order) | Europe is a continent located entirely in the Northern Hemisphere and mostly in the Eastern Hemisphere. |
| $x_2$ | Europe is a continent located entirely in the Northern Hemisphere and mostly in the Eastern Hemisphere. Spain is a member of the European Union. |
| $x_3$ | Europe is a continent located entirely in the Northern Hemisphere and mostly in the Eastern Hemisphere. France is a member of the European Union. |
| $x_4$ | Europe is a continent located entirely in the Northern Hemisphere and mostly in the Eastern Hemisphere. France is is a lieing country in the world. It is a bunch of crap. |
| $x_5$ | Europe is a continent located entirely in the ~~Northern~~ Hemisphere and mostly in the Eastern Hemisphere. ~~France is a lieing country in the world. It is a bunch of crap.~~ There is a type of debate of a group of people who are not considered to be a part of the United Nations. |
| Initial Sentence (2-order) | Europe is a continent located entirely in the Northern Hemisphere and mostly in the Eastern Hemisphere. |
| $x_2$ | Europe is a continent located entirely in the Northern Hemisphere and mostly in the Eastern Hemisphere. The Western South Eastman Islands are also located in Europe. |
| $x_3$ | Europe is .k.ka.j.jf.go.skxklse |
| $x_4$ | Europe is ~~.k.ka.j.jf.go.skxklse~~ a continent in the Northern Hemisphere. The Islands are also in Europe and they are great. |

Table 4: Example generation when sampling with an edit model. We notice that the 2nd order model is able perform a revert operation given the context fed through the edit-compressed sequence about the previous revision, whereas the 1-order model although deleting its generated spam, generates something relatively unrelated. However we note that this reversion is not exact (likely due to the information loss during edit compression). This corresponds with our observations in our qualitative study (where likelihood of reverted edits is increased in the 2+ order models).

contrast with this work, these approaches only consider modeling single editing steps instead of the more general multi-step setting tackled here.

**Semi-Autoregressive Sequence Generation.** Work in machine translation has explored non-autoregressive methods that use an iterative generation process. This was first proposed by Lee et al. (2018) using latent variables and denoising autoencoding. Iterative refinement was then explored with conditional masked language models (CMLM; Ghazvininejad et al., 2019), simplifying previously proposed methods, by iteratively replacing predicted tokens with low confidence. Gu et al. (2019), introduced the Levenshtein Transformer, making this approach more flexible by introducing insertion and deletion operations. However, these methods have not yet considered modeling *natural* editing processes, instead using either random sampling or heuristically determined orders.

**Other Editing-based Work.** Other work on editing has included editing for sentence fusion (Malmi et al., 2019), in which one can perform minimal operations to join two sentences together grammatically. Furthermore, with text style transfer in which the difference between sentences in different styles (e.g. positive vs negative) can be relatively minimal (Reid & Zhong, 2021; Mallinson et al., 2020). Furthermore, Faltings et al. (2021) explored natural language conditioned editing as a means for controllable text generation using a T5-based (Raffel et al., 2020) sequence to sequence model. Also related to our work is text morphing (Huang et al., 2018), in which they look at an edit-based model to interpolate between two sentences. We also note that diffusion models (Sohl-Dickstein et al., 2015; Ho et al., 2020) can be formulated as a flavor of editing models, where the model learns to iteratively edit some representation of information in order to construct a final version of said representation.

## 8 CONCLUSIONS

In this work, we proposed the novel task of modeling editing processes, in which we model the likelihood of documents by way of their natural editing processes. We develop new datasets and curate existing datasets for downstream tasks. We find that modeling editing processes is beneficial to this end, in contrast to modeling single-order edits, as has been done in much of previous literature. More broadly, we believe that tackling iterative refinement tasks in broader NLP may be beneficial given its resemblance to the natural generative and creative process. In future work, we look to investigate methods for transferring pre-trained edit models to a wider range of NLP tasks.

## ETHICS STATEMENT

This work has impact in the fields of natural language processing and machine learning, and proposes a new task of modeling editing processes, while proposing new datasets and models to this end. There are several potential ethical concerns. First, we initialize our models with a pre-trained model (trained on webtext), which may capture and amplify biases found in the pre-training data. We further fine tune on natural data which, in addition to benign edits, also contains a portion of human-generated spam and potentially offensive edits (especially in the natural language case), which may also amplify and capture biases in that regard. On the other hand, we have also demonstrated the ability of the proposed model to identify and revert adverse edits performed by spammers or malicious editors, a potentially useful tool for content moderation, or even detecting automatically generated spam edits, as has been done in the detection of fake news (Zellers et al., 2019).

## REPRODUCIBILITY STATEMENT

As noted on the first page, all code and datasets necessary to reproduce the experiments in this paper will be released upon conclusion of paper review. Furthermore, all data processing steps are discussed in Section 4 and Appendix A.1 for both code and natural language.

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

## A APPENDIX

### A.1 FURTHER TRAINING DETAILS

When training, we employ data sharding to enable cheaper, on the fly data processing. We shard each documents' into 10 shards and form splits based on these shards. Our train-valid-test splits are split 90%,5%,5% for commit message generation, commit-conditioned edit modeling, and edit modeling. We use a dropout value of 0.3 and use the GELU activation for all MLPs. We use a learning rate of 1e-4 warmed up for 1000 iterations. We also note that we extended the positional embedding matrix for BART to handle longer sequences.

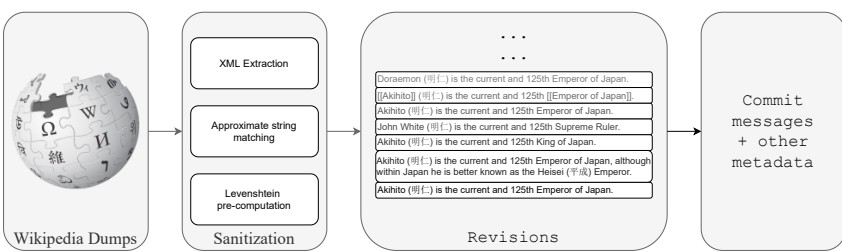

Figure 3: An overview of the WIKIREVISIONS data generation process for collecting clean multi-step revision data.

## A.2 ABLATION STUDY ON THE IMPACT OF COMMENTS ON COMMENT-CONDITIONED EDIT MODELING

| Model | ePPL ($\Delta$ compared to 2-order) |
|---|---|
| EDITPRO (2-order) | 53.91 (–) |
| EDITPRO (2-order, only current revision is comment conditioned) | 52.06 (-1.85) |
| EDITPRO (2-order, completely comment-conditioned) | 51.83 (-2.08) |
| EDITPRO (3-order) | 50.84 (-3.07) |

Table 5: Ablation on the impact of comments in 3-order edit modeling

We include an additional experiment, demonstrating the impact of including comment information in previous revisions, and not including comment information in comment conditioned edit modeling. We do this by including a 2-order edit model trained with only the current revision being comment-conditioned. The results demonstrate that a majority of the performance gain is indeed derived from modeling an extra revision, with the clear outperformance of 3-order edit modeling versus 2-order edit modeling. We also note that there is some gain that is derived from modeling previous revisions to be also comment conditioned, however we note that this is relatively minimal compared to the comment conditioning of the current revision and modeling an extra order.

