# OpenReview forum: "Learning to Model Editing Processes"
_ICLR.cc/2022/Conference — ICLR 2022 Submitted_

### Official Review · Reviewer_DCPz · 2021-10-26

**Correctness:** 2
**Technical Novelty And Significance:** 3
**Empirical Novelty And Significance:** 3
**Recommendation:** 5
**Confidence:** 4

**Main Review:**

The outset of the paper is valid: Conditioning on the complete (or longer) revision history can potentially model the editing process of documents more accurately than models that just take the latest history into account. However, I'm not completely convinced by the evaluation. I think that it is hard to justify an approach with perplexity improvements only. Perplexity can be a valuable tool to get a better understanding of the system, but ultimately showing the usefulness in a downstream task can make a much stronger case. The two downstream tasks in the paper that do not use perplexity (Tab.3 / Sec. 6.2) are (a) predicting code commit messages / Wikipedia edit comments and (b) semantic intent classification. If I understand correctly, both of these tasks actually do not require the *prediction* of the document edits. Tab. 3 does not compare with related work on these tasks. BLEU and F1 scores do improve with increasing order which is a good sign, but simple baselines/ablations would strengthen the results. For example, task (a) may simply benefit from having access to previous commit messages, not from a better modeling of the edits. Similarly, for (b) the model could learn that two subsequent "Vandalism" revisions are unlikely without using edits from previous revisions.

Minor comments:
- The paper sometimes claims that previous models are just able to represent single *edits* (abstract, Sec. 4). This is, of course, not true: they are able to represent multiple edits with a document, just in a single *edit step*. This subtle difference is quite important so the wording should be more precise.
- Eq. 2: This might be true in theory, but it is highly impractical since it seems unlikely that the sum over all document revision histories can be approximated to a reasonable degree, let alone enumerated.
- Eq. 2: The notation under the sigma is wrong. It should start with \tilde{X}\in\{...\}
- Sec. 2: "x_0 represents [...] generally the null string". Is this true for both WikiRevisions and CodeRevisions tasks? Doesn't this significantly increase the length of the edit representations for subsequent revisions? x_0 is not empty in Fig. 1.


**Summary Of The Paper:**

Existing edit-based models condition on a single sentence or document. This paper argues that such frameworks do not represent processes well in which the same document undergoes multiple iterations of revisions, for example the edit histories of Wikipedia articles or source code. The proposed system is a two-stage model that first labels each token in the next revision with one of four edit labels, and then employs a seq2seq second stage model to predict the output tokens for generative labels such as INSERT and REPLACE. Both stages do not only condition on the previous revision but on the n latest revisions. Gains in perplexity are shown, as well as improvements of two downstream tasks: "edit-conditioned generation" (generating meta information to edits such as comments) and "edit-conditioned classification" (1-way semantic intent classification).

**Summary Of The Review:**

While there are some potential merits to the idea of modeling multiple revisions, I have doubts whether the evaluation demonstrates them in a convincing way.

---

> ### Author Response · Authors · 2021-11-23
> **Response**
>
> We thank the reviewer for their helpful feedback. We respond to your concerns below and hope that through discussion, we can satisfy you that the current work provides the groundwork and lays the concept of modelling edit processes enough to warrant publication. We hope that afterwards you will consider revising your assessment. Thank you very much!
>
> 1) Table 3 does not compare with related work on these tasks.
> - The aim of Table 3, was to compare the impact of adding various revisions when performing edit-related downstream tasks. Previous models have not been shown to handle this, hence our lack of comparison in this regard.
>
> ---
>
> 2) If I understand correctly, both of these tasks actually do not require the prediction of the document edits. For example, task (a) may simply benefit from having access to previous commit messages, not from a better modeling of the edits. Similarly, for (b) the model could learn that two subsequent "Vandalism" revisions are unlikely without using edits from previous revisions.
>
> - This is a great point! We are looking for edit-prediction tasks to test our models perform on downstream tasks in future work --- one shortcoming of this is that many of these tasks in existing literature do not have multi-step revisions. Given this, we would have to formulate new downstream tasks which take into account multiple revisions, which albeit interesting and most definitely important is out of the scope of this work. However, in this case, we wanted to test how modelling editing processes versus single-steps matter with other tasks (to complement our proposed edit modeling task---a prediction task) that are intrinsically edit-based.
>
>     Regarding the scenarios mentioned, however: for task (a), given the results of Table 2, the mentioned scenario is unlikely as purely modelling the edits do improve ePPL and a 3-order model outperforms a 2-order comment conditioned model / 2-order model outperforms a 1-order comment conditioned model. Furthermore, for task (b) if the model is able to grasp this (note that labels for previous revisions are not fed to the model), then this is a positive sign that the model is learning to model the connection between edits to a certain degree. (edit: we include an additional ablation experiment in appendix A.2 to quantify this notion for task (a))
>
> ---
>
> 3) Sec. 2: "x_0 represents [...] generally the null string". Is this true for both WikiRevisions and CodeRevisions tasks? Doesn't this significantly increase the length of the edit representations for subsequent revisions? x_0 is not empty in Fig. 1.
>
> - Yes, starting from the null string is the case for all edit modeling datasets. Given that we start from a null string, when modelling, the first step is always to general the initial version (x_1) of the document. The length of the edit representations are compressed with our proposed approach using the edit operations, so document length does not directly correlate with edit representation length in this context. We ~~will change~~ have changed the figure to reflect this.

---

### Official Review · Reviewer_EhJP · 2021-11-01

**Correctness:** 2
**Technical Novelty And Significance:** 3
**Empirical Novelty And Significance:** 2
**Recommendation:** 5
**Confidence:** 4

**Main Review:**

Strengths:
 - Modeling the task of sequence editing as a multistep, iterative process. Almost all previous methods model sequence editing as a single-step process.  The proposed method is technically sound.

 - Edit processes are modeled on a larger scale e.g. document-level edits.

 - Modeling multi-order edits up to the order of 3, does result in improved performance (but only in comparison to the same model with an edit order of 1)

 - New proposals for datasets and metrics related to iterative sequence editing: Edit Perplexity, Generation Perplexity, and Operation Perplexity. (However, the model's design explicitly favors these metrics.)

Limitations / Concerns

- Comparisons with single-step edit models: There are prior works on iterative sequence editing that utilize single-step edit models and find them useful. E.g. see Table 4 in [1]. How effective is the proposed model in comparison to these methods?

 - The paper starts with the motivation that "Revising and editing are a central part of the human creative workflow, with most original content being developed not in a single iteration, but in many iterations with each more refined than the last". I agree with the authors on this part. However, the datasets considered in this paper may not be well aligned with this motivation. E.g. in many cases, Wikipedia edits and code edits may not be just stylistic edits required for improving sentence formation. A good portion of edits might also introduce new information as well.

 - Grammatical Error Correction is a very well-known task in the domain of sequence editing and aligned with the motivation of iteratively refining and editing a given sentence. The experimental observations could have been more convincing if the authors could show that their method is more effective than prior single-step edit methods for such tasks.

 - The proposed evaluation metrics in Section 5.2, are very much tied to the model design (Eqn 6). Thus the proposed metrics may naturally favor the model over other models which were not trained using similar objectives. Hence, I feel that the proposed model requires a more rigorous evaluation.

 - The proposed method is only compared with a single baseline. In the related work, authors do acknowledge prior work on sequence editing. However, a comparison with those methods is currently not provided.

 - Details about the posterior regularization being used in Section 6.2 are not fully clear.

 - It would be interesting to know how important is the order of edit modeling, in Table 2 and Table 3? Does performance gains diminish beyond order 3?


References:

[1] Parallel Iterative Edit Models for Local Sequence Transduction (https://aclanthology.org/D19-1435.pdf)



**Summary Of The Paper:**

This paper considers the task of explicitly modeling the process of editing sequences in an iterative manner, while the prior approaches for sequence editing are usually single-step editing methods. Thus, the proposed method is significantly novel. The proposal is to decompose the editing process into (i) predicting edit operations (insert, substitute, delete, and keep) and (ii) generating spans corresponding to those operations. Both these steps are conditioned on the previous versions of the input sequence and its edit history. The model first predicts edit operations conditioned on previous documents and then revises the document conditioned on edits and previous documents.

In my opinion, the experimental section can be improved significantly. It currently lacks sufficient comparisons with prior work to demonstrate the efficacy of iterative edit modeling over prior methods of single-step editing. The proposed evaluation metrics also seem to be directly aligned with the design of the proposed model, hence comparisons with other models on these metrics might be unfair.

**Summary Of The Review:**

This paper investigates an interesting problem of modeling the iterative process of editing the sequences. The ideas proposed seem to be technically sound and novel.

However, I am not convinced with the tasks, metrics, and comparisons with prior work on sequence editing.

Metrics: In my opinion, metrics seem to be directly aligned with the model's design, and hence comparing with models which do not optimize these metrics could be unfair.

Tasks: Tasks like Grammatical Error Correction or paraphrasing are widely known and directly aligned with the paper's motivation of iteratively editing. The effectiveness of the proposed method on such tasks can make experiments significantly more convincing.

Comparisons with prior work: The paper compares with only one baseline, while it does acknowledge well-known and recent papers in the domain of sequence editing. The paper rightly argues that most of the prior work models sequence editing as a single-step process. However, the significance of modeling edits in multiple iterative steps will be more evident after comparisons with single-step edit models.

---

> ### Author Response · Authors · 2021-11-23
> **Response**
>
> Thank you for taking the time to give this feedback. We are glad you find the task/ideas to be novel and interesting. We respond to your concerns below with regard to the experiments in the section below and hope that, following the points of clarification highlighted by the reviewer, we can resolve any ambiguities in the paper that might hinder any impact, and thank you for helping make the paper stronger as a result. We look forward to hearing your thoughts on our response and hope that you will be willing to revise your score after discussion.
>
>
> 1) Comparisons with single-step edit models: There are prior works on iterative sequence editing that utilize single-step edit models and find them useful. E.g. see Table 4 in [1]. How effective is the proposed model in comparison to these methods?
>
> - This is precisely what we tested with EditPro (1 order) and LEWIS models. However, we demonstrate with extra context (2,3-order) these refinements models do much better from a likelihood prediction perspective. As far as we know, other single-step models are quite similar in mechanism to LEWIS, and LEWIS reports state-of-the-art results on the benchmarks it is tested on, but we would be happy to consider any other baselines that you think are relevant.
>
> ---
>
> 2) The paper starts with the motivation that "Revising and editing are a central part of the human creative workflow, with most original content being developed not in a single iteration, but in many iterations with each more refined than the last". I agree with the authors on this part. However, the datasets considered in this paper may not be well aligned with this motivation. E.g. in many cases, Wikipedia edits and code edits may not be just stylistic edits required for improving sentence formation. A good portion of edits might also introduce new information as well.
>
> - We do agree with this point, but that is also our point as well! We note that introducing new information is an essential part of the creative workflow, and is a given especially when starting from a blank page (or null string).
>
> ---
>
> 3) Grammatical Error Correction is a very well-known task in the domain of sequence editing and is aligned with the motivation of iteratively refining and editing a given sentence. The experimental observations could have been more convincing if the authors could show that their method is more effective than prior single-step edit methods for such tasks.
> - We agree with this point, however, with GEC there is no available multi-step edit data. If there is and we are missing it we would be happy to test it out.
>
> ---
>
> 4) The proposed evaluation metrics in Section 5.2, are very much tied to the model design (Eqn 6). Thus the proposed metrics may naturally favor the model over other models which were not trained using similar objectives. Hence, I feel that the proposed model requires a more rigorous evaluation.
> - We would like to note that the LEWIS baseline model was trained with the exact same objective, however, we are open to suggestions on other more rigorous evaluation metrics that nonetheless capture the essence of our proposed task of modeling editing processes.
>
> ---
>
> 5) The proposed method is only compared with a single baseline. In the related work, authors do acknowledge prior work on sequence editing. However, a comparison with those methods is currently not provided.
> - Many prior methods use editing style methodology for other tasks, such as machine translation, style transfer, etc… Many of these other baselines employ similar methods (e.g. LEWIS, Levenshtein Transformer) or use a sequence-to-sequence model as an editor. Given that full sequence to sequence models (1) generally learn to copy most of the content (when the sequence length is long), and (2) are computationally expensive (using a full generation process to copy the sequence), we compare with a recent state-of-the-art baseline: LEWIS. However, note that our goal in this paper is not to propose a new model architecture, but to introduce the new task of modelling editing processes and methodology to learn multi-order modelling of edits. In fact, some concepts introduced here regarding multi-order editing could also be incorporated into other architectures as well.
>
> ---
>
> 6) Details about the posterior regularization being used in Section 6.2 are not fully clear.
> - Given the model can get overconfident with “KEEP” scores, making it hard to attain diverse samples, at inference we regularize the probability (simply dividing the KEEP probability by a given factor) until it roughly matches the distribution found in the training set.
>
> ---
>
> 7) It would be interesting to know how important is the order of edit modeling, in Table 2 and Table 3? Does performance gains diminish beyond order 3?
> - We expect performance gains to diminish past order 3 (as we already have diminishing returns for 1->2, and 2->3 order models), however, we did not perform these experiments due to GPU memory limitations.

---

### Official Review · Reviewer_eci3 · 2021-11-02

**Correctness:** 3
**Technical Novelty And Significance:** 1
**Empirical Novelty And Significance:** 2
**Recommendation:** 3
**Confidence:** 4

**Main Review:**

Strengths:
1. The idea is simple and intuitive
2. The paper is well organized and easy to follow

Weaknesses:
1. My main concern is the motivation of this work. It is not clear why interactively editing is better than single-step editing. The authors need to elaborate on the source of the gain in more detail.
2. Similar idea (interactively editing) is already presented in work[1]. The authors do not take this method as a baseline, or at least they should thoroughly discuss the differences with this work.
3. In section 6, the paper only compares their model to a limited baseline, making the experiment less convincing. The paper needs to add some baseline, especially in the dataset CODEREVISIONS.

References: [1] Shaohan, Yu, Furu, Ming (2018). Text Morphing.

**Summary Of The Paper:**

This paper proposes a new task of modeling editing processes to model the whole process of iteratively generating sequences. To tackle the new challenge, the authors propose a conceptual framework to describe the likelihood of multi-step edits and describe neural models that can learn a generative model of sequences based on these multi-step edits. The Experimental results show that modeling editing processes improve performance compared to previous single-step models of edits on related downstream tasks and the proposed task.

**Summary Of The Review:**

Please refer to the Weaknesses in Main Review, I hope the authors could give a clear answer

---

> ### Author Response · Authors · 2021-11-23
> **Response**
>
> We thank the reviewer for their comments and are glad that they appreciated the presentation and the idea.  We hope concerns are addressed in our response below and hope that you will update your score in light of this:
>
> 1) My main concern is the motivation of this work. It is not clear why interactively editing is better than single-step editing. The authors need to elaborate on the source of the gain in more detail.
> - Many real-world editing processes (such as those done by human Wikipedia and Github authors) are multi-step. Current single-step methods are equivalent to writing a document by only making a single change at a time, forgetting how and why you came to that point, and repeating the process. Intuitively this is unsatisfying as it’s obviously not how human editors generate documents, and empirically we also demonstrate accuracy gains both on edit modelling itself and other downstream tasks in the experiments. We believe this is relatively clearly described in the intro and throughout the paper, but we would be happy to have any suggestions for improvement in the exposition.
>
> ---
>
> 2) Similar idea (interactively editing) is already presented in work[1]. The authors do not take this method as a baseline, or at least they should thoroughly discuss the differences with this work.
>
> - Thank you for sharing this work! We were not aware of this but will be sure to cite and compare with it. However, [1] presents a model that is 1) not tested on modelling natural edit processes, and 2) is used largely for interpolating between sentences with by way of discrete edit operations, making their work orthogonal to ours.
>
> ---
>
> 3) In section 6, the paper only compares their model to a limited baseline, making the experiment less convincing. The paper needs to add some baseline, especially in the dataset CODEREVISIONS.
>
> - We did not add the LEWIS baseline for CodeRevisions, as it uses pre-trained BART and RoBERTa models which we did not use for code (given the different domain). However, does the reviewer have any other baselines they recommend we test?

---

### Official Review · Reviewer_vM6c · 2021-11-04

**Correctness:** 2
**Technical Novelty And Significance:** 2
**Empirical Novelty And Significance:** 3
**Recommendation:** 5
**Confidence:** 5

**Main Review:**

Strong points of the paper:
1. Previous approaches only consider predicting editing based on the current text (therefore just one revision history). This paper considers more revision to predict and it show more revisions (up to three) are beneficial.
2. The paper prepares two corpus with full revision history, which might be beneficial for the community on future research.
3. The paper propose a modified semi-autoregressive Transformer model to predict both the editing operator and generator for insertion and replacement operations. The model is reasonable.
4. The paper shows improvement on the metrics on WikiRevisions dataset.

Weak points of the paper:
1. One of the metrics proposed in the paper is slightly strange. Why to measure the perplexity of operation? Why not measuring the accuracy or F1 score of the per-token (excluding the no-edit operation).
2. There are some details missing about the method in Section 3.2. What is the network for autoregressive operation prediction $P(e_j | e_1^{j-1})$? What is the decoder for insertion and replacement? Is it Transformer decoder?
3. Examples in Table 4 actually show that this EditPro model generates rather bad text. Those text are fluent but semantically incoherent.
4. continue from the above point, Human evaluation of the editing quality is missing. Do you conduct human evaluation of the generated text?
5. It seems a rather simple baseline is missing. If you directly use Transformer encoder-decoder to predict both the operation and the generation text, how is EditPro compare to this baseline?

It could be a good paper if the authors could fix some major concerns above.

Some minor issue:
It seems the perplexity equations are incorrect, missing negative sign.

There are also additional related work on editing based method for text generation. [1] uses simulated annealing to search for edit operations, and [2] uses Metropolis-Hastings sampling for editing. Do those apply to the edit generation problem here?

[1]: Unsupervised Text Generation by Learning from Search. 2020.

[2]: CGMH: Constrained Sentence Generation by Metropolis-Hastings Sampling. 2019

**Summary Of The Paper:**

The paper presents a new problem of modelling the series editing operations of an article, i.e. $p(x_1 \dots x_n )=\prod_t p(x_t | x_1\dots x_{t-1}) $, where each $x_t$ represents an article after t-th editing. There are three types of operations: insert, delete, and replacement. The paper propose a model based on a modified Transformer to predict both the edit operation and the content to be generated for insertion and replacement. To predict the edit operation type (including additional no-edit action), the model uses a Transformer encoder to encoder the k previous version of history (if $k=1$, it is just based previous version) and a simple autoregressive MLP to predict the type of editing operation for each of the token. For consecutive insertion and replacement edits, the model will take the average contextual representation of the span to generate tokens (together with the span start and end index and edit type embedding).
For evaluation, the paper constructs two datasets with editing history: WikiRevisions (from wikipedia editing history with some cleaning) and CodeRevisions (from 700 python repositories on Github with MIT license, over 1000 commits and 500 stars).
The paper propose a set of three metrics: operation perplexity, generation perplexity (given operation), and the editing perplexity combining both. On these metrics, the paper shows the proposed EditPro model improves over baseline LEWIS method on WikiRevisions data. It also shows more edit history is helpful in prediction, therefore EditPro based on three previous revision history is better than that based on one immediate previous version.

The paper also propose three downstream tasks for evaluation: editing given the text comment (from the commit message), edit-conditioned generation, and edit-condition classification.


**Summary Of The Review:**

Reason to accept: new problem setup for editing based text generation, two new datasets, and evidence of multiple history helps generation.

Reasons to reject: one metric is questionable, examples showing the method not working, missing human evaluation and a straightforward baseline. Without those, it is hard to judge the quality of the generation.

---

> ### Author Response · Authors · 2021-11-23
> **Response**
>
> We thank the reviewer for their time and detailed feedback. We respond to the points you make below and hope we clarify most of your concerns. We look forward to your response and hope you will consider revising your assessment in light of this process. Thank you very much!
>
>
> 1) One of the metrics proposed in the paper is slightly strange. Why to measure the perplexity of operation? Why not measuring the accuracy or F1 score of the per-token (excluding the no-edit operation).
>
> - One thing to note is that because this is an unconditional language modeling task, there are many many potential valid next edits that could be done at any particular time. Because of this, even for a good model the accuracy of predicting a particular next edit may be very low. That is why we opt for perplexity, which is discriminative with respect to the quality of the model even within this setting.
>
> ---
>
> 2) There are some details missing about the method in Section 3.2. What is the network for autoregressive operation prediction? What is the decoder for insertion and replacement? Is it Transformer decoder?
>
> - For insertion and replacement, we use a transformer decoder conditioned on the vector representation of the aggregated span (or the previous token in the case of replacement) and the target operation. For autoregressive operation prediction, we also use a transformer decoder (using a causal mask). (This is mentioned in Section 3.2, but we will clarify this in revision)
>
> ---
>
> 3) Examples in Table 4 actually show that this EditPro model generates rather bad text. Those text are fluent but semantically incoherent.
> - We agree that the qualitative examples are not perfectly semantically coherent (although they do show topical coherence and some degree of discourse coherence). The main point of the example was to demonstrate that the context-aware modeling of editing processes (the main contribution of our paper) improves the ability of the model to decide what and what kind of edits to perform next, which these examples show through the improved ability of the model to revert previous edits. Note that this is a relatively small language model, trained from scratch on only Wikipedia data with relatively complex edits, and we expect that if we scaled up the model further in terms of size or training data the semantic coherence of the generated text would improve. We hope that the proposed new task of modeling editing processes, as well as the provided data and first models are useful nonetheless. We will add some more discussion of this in the revised version of the paper. (edit: we have added discussion of this, doing our best given the page constraints)
>
> ---
>
> 4) Human evaluation of the editing quality is missing. Do you conduct human evaluation of the generated text?
> - Thank you for the comment! We do believe this could be useful and plan on adding this in a future revision, however could not do so in the response period due to lack of time. We also would like to add a more systematic qualitative analysis of the results, which would help discuss the issues of semantic coherence mentioned above as well.
>
> ---
>
> 5) It seems a rather simple baseline is missing. If you directly use Transformer encoder-decoder to predict both the operation and the generation text, how is EditPro compare to this baseline?
>
> - This is quite similar to LEWIS and our single-order baseline. However using a pure sequence-to-sequence model is not able to edit at inference time for long sequences, given its tendency to learn to copy the great majority of the tokens.
>
> ---
>
> 6) There are also additional related work on editing based method for text generation. [1] uses simulated annealing to search for edit operations, and [2] uses Metropolis-Hastings sampling for editing. Do those apply to the edit generation problem here?
>
> - These approaches are both interesting, and we can particularly see them being used to adapting pre-trained edit models to various downstream tasks which require editing. However, in this case, we model natural editing processes without specific external constraints.

---

> > ### Comment · Reviewer_vM6c · 2021-11-27
> > **Thanks for your clarification!**
> >
> > Your comments do clarify the paper. I have a better understanding of the paper. There is still remaining question about the quality evaluation. It would be much supportive if you could provide human evaluation comparing the quality of these methods.

---

### Decision · Program_Chairs · 2022-01-20

**Decision:**

Reject

**Comment:**

This paper introduces a novel task (i.e., modelling the iterative process of editing sequences) and proposes a Transformer-based architecture to address it tractably. The paper also elects a number of metrics that are argued to shed enough light onto the merits of the proposed architecture.

In our view, the current version is not ready for acceptance. Here are some of the reasons I'd highlight:

* It is not entirely clear to us that the task in consideration has enough substance to grant acceptance nor that it speaks to a large enough audience. Perhaps the challenges identified here are more general and the developments for this task can be extended to related generation problems? If so, this is something one could consider for a revised version of the paper.
* The motivation does not seem to align well with the datasets used to demonstrate the task. Perhaps the difficulty to find a dataset that matches the motivation is an indication that the task and its challenges are tad too specific.
* It's the impression of more or less everyone involved that the paper lacks comparisons, and that the evaluation is not thorough enough, and the rebuttal did not ease our concerns sufficiently.